# PG-TS: Improved Thompson Sampling for Logistic Contextual Bandits

**Bianca Dumitrascu**[*]
Lewis Sigler Institute for Integrative Genomics
Princeton University
Princeton, NJ 08540
biancad@princeton.edu

**Karen Feng**[*]
Department of Computer Science
Princeton University
Princeton, NJ 08540
karenfeng@princeton.edu

**Barbara E Engelhardt**
Department of Computer Science
Princeton University
Princeton, NJ 08540
bee@princeton.edu

## Abstract

We address the problem of regret minimization in logistic contextual bandits, where a learner decides among sequential actions or arms given their respective contexts to maximize binary rewards. Using a fast inference procedure with Pólya-Gamma distributed augmentation variables, we propose an improved version of Thompson Sampling, a Bayesian formulation of contextual bandits with near-optimal performance. Our approach, Pólya-Gamma augmented Thompson Sampling (PG-TS), achieves state-of-the-art performance on simulated and real data. PG-TS *explores* the action space efficiently and *exploits* high-reward arms, quickly converging to solutions of low regret. Its explicit estimation of the posterior distribution of the context feature covariance leads to substantial empirical gains over approximate approaches. PG-TS is the first approach to demonstrate the benefits of Pólya-Gamma augmentation in bandits and to propose an efficient Gibbs sampler for approximating the analytically unsolvable integral of logistic contextual bandits.

## 1   Introduction

A contextual bandit is an online learning framework for modeling sequential decision-making problems. Contextual bandits have been applied to problems ranging from advertising [1] and recommendations [22, 21] to clinical trials [37] and mobile health [33]. In a contextual bandit algorithm, a learner is given a choice among $K$ actions or arms, for which contexts are available as $d$-dimensional feature vectors, across $T$ sequential rounds. During each round, the learner uses information from previous rounds to estimate associations between contexts and rewards. The learner's goal in each round is to select the arm that minimizes the cumulative regret, which is the difference between the optimal oracle rewards and the observed rewards from the chosen arms. To do this, the learner must balance *exploring* arms that improve the expected reward estimates and *exploiting* the current expected reward estimates to select arms with the largest expected reward. In this work, we focus on scenarios with binary rewards.

To address the exploration-exploitation trade-off in sequential decision making, two directions are generally considered: Upper Confidence Bound algorithms (UCB) and Thompson Sampling (TS).

---

[*]indicates equal authorship

UCB algorithms are based on the principle of optimism in the face of adversity [3, 6, 15] and rely on choosing actions according to expected rewards perturbed by their respective upper confidence bounds. Based on Bayesian ideas, TS [34] assumes a *prior* distribution over the parameters governing the relationship between contexts and rewards. At each step, an action corresponding to a random parameter sampled from the posterior distribution is chosen. Upon observing the reward for each round, the posterior distribution is updated via Bayes' rule. TS has been successfully applied in a wide range of settings [2, 32, 9, 28].

While UCB algorithms have simple implementations and good theoretical regret bounds [22], TS achieves better empirical performance in many simulated and real-world settings without sacrificing simplicity [9, 15]. Furthermore, TS is amenable to scaling through hashing, thus making it attractive for large scale applications [20]. In addition, recent studies have bridged the theoretical gap between TS and UCB based methods by analyzing regret and Bayesian regret in TS approaches for both generalized linear bandits and reinforcement learning settings [2, 28, 26, 29, 4, 5].

In this work, we focus on improving the TS approach for contextual bandits with logistic rewards [9, 15]. The logistic rewards setting is of pragmatic interest because of its natural application to problems such as modeling click-through rates in advertisement applications [22]. Computationally, the functional form of its logistic regression likelihood leads to an intractable posterior – the necessary integrals are not available in closed form and difficult to approximate. This intractability makes the sampling step of TS with binary or categorical rewards challenging. From an optimization perspective, the logistic loss is exp-concave, thus allowing second-order methods in a purely online setting [19, 25]. However, the convergence rate is exponential in the number of features $d$, making these solutions impractical in most real-world settings [19].

Existing Bayesian solutions to logistic contextual bandits rely on regularized logistic regression with batch updates in which the posterior distribution is estimated via Laplace approximations. The Laplace approximation is a second-order moment matching method that estimates the posterior with a multivariate Gaussian distribution. Despite offering asymptotic convergence guarantees under restricted assumptions [7], the Laplace approximation struggles when the dimension of the context (arm features) is larger than the number of arms, and when the features themselves are non-Gaussian. Both of these situations arise in the online learning setting, creating a need for novel TS approaches to inference. Recent work suggests that a double sampling approach via MCMC can improve TS [35]. This approach provides MCMC schemes for bandits with binary and Gaussian rewards, but these algorithms do not generalize to the logistic contextual bandit.

We propose Pólya-Gamma augmented Thompson sampling (PG-TS), a fully Bayesian alternative to Laplace-TS. PG-TS uses a Gibbs sampler built on parameter augmentation with a Pólya-Gamma distribution [27, 36, 31]. We compare results from PG-TS to state-of-the-art approaches on simulations that include toy models with specified and unspecified priors, and on two data sets previously considered in the contextual bandit literature.

The remainder of this paper is organized as follows. Section 2 reviews relevant background and introduces the problem. The details of Pólya-Gamma augmentation are provided in Section 3. Section 4 includes an empirical evaluation and shows substantial performance improvements in favor of PG-TS over existing approaches. We conclude in Section 5.

## 2  Background

In the following, $\mathbf{x} \in \mathbb{R}^d$ denotes a $d$-dimensional column vector with scalar entries $x_j$, indexed by integers $j = \{1, 2 \ldots d\}$; $\mathbf{x}^\top$ is transposed vector $\mathbf{x}$. $\mathbf{X}$ denotes a square matrix, while $X$ refers to a random variable. We use $\| \cdot \|$ for the 2-norm, while $\|\mathbf{x}\|_{\mathbf{A}}$ denotes $\mathbf{x}^\top \mathbf{A} \mathbf{x}$, for a matrix $\mathbf{A}$. Let $\mathbb{1}_{\mathcal{B}}(x)$ be the indicator function of a set $\mathcal{B}$ defined as 1 if $x \in \mathcal{B}$, and 0 otherwise. $MVN(\mathbf{b}, \mathbf{B})$ denotes a multivariate normal distribution with mean $\mathbf{b}$ and covariance $\mathbf{B}$, and $\mathbf{I}_d$ is the $d \times d$ identity matrix.

### 2.1  Contextual Bandits with Binary Rewards

We consider contextual bandits with binary rewards with a finite, but possibly large, number of arms $K$. These models belong to the class of generalized linear bandits with binary rewards [15]. Let $\mathcal{A}$ be the set of arms. At each time step $t$, the learner observes contexts $\mathbf{x}_{t,a} \in \mathbb{R}^d$, where $d$ is the number of features per arm. The learner then chooses an arm $a_t$ and receives a reward $r_t \in \{0, 1\}$.

The expectation of this reward is related to the context through a parameter $\boldsymbol{\theta}^* \in \mathbb{R}^d$ and a logistic link function $\mu$: $E[r|\mathbf{x}] = \mu(\mathbf{x}^\top \boldsymbol{\theta}^*)$, where $\mu(z) = \exp(z)/(1 + \exp(z))$.

For example, in a news article recommendation setting, the recommendation algorithm (*learner*) has access to a discrete number of news articles (*arms*) $\mathcal{A}$ and interacts with users across discrete trials $t = 1, 2, \ldots$ where the logistic reward is whether or not the user clicks on the recommended article. The articles and the users are characterized by attributes (*context*), such as genre and popularity (*articles*), or age and gender (*users*). At trial $t$, the learner observes the current user $u_t$, the available articles $a \in \mathcal{A}$, and the corresponding contexts $\mathbf{x}_{t,a}$. The context is a $d$-dimensional summary of both the user's and the available articles' context. At each time point, the goal of the learner is to provide the user with an article recommendation (arm choice) that they then may choose to click (reward of 1) or not (reward of 0). The relationship between rewards and contexts is mediated through an underlying coefficient vector $\boldsymbol{\theta}^*$, which can be interpreted as an encoding of the users' preferences with respect to the various context features of the articles.

Formally, let $\mathcal{D}_t$ be the set of triplets $(\mathbf{x}_{i,a_i}, a_i, r_i)$ for $i = 1, \ldots, t$ representing the past $t$ observations of the contexts, the actions chosen, and their corresponding rewards. The objective of the learner is to minimize the cumulative regret given $\mathcal{D}_{t-1}$ after a fixed budget of $t$ steps. The regret is the expected difference between the optimal reward received by always playing the optimal arm $a^*$ and the reward received following the actual arm choices made by the learner.

$$r_t = \sum_{i=1}^{t} \left[ \mu(\mathbf{x}_{i,a^*}^\top \boldsymbol{\theta}^*) - \mu(\mathbf{x}_{i,a_i}^\top \boldsymbol{\theta}^*) \right] \tag{1}$$

The parameter $\boldsymbol{\theta}$ is reestimated after each round $t$ using a generalized linear model estimator [15], The point estimate of the coefficient at round $t$, $\boldsymbol{\theta}_t$, can be computed using approaches for online convex optimization [18, 19]. However, these approaches scale exponentially with the context dimension $d$, leading to computationally intractable solutions for many real world contextual logistic bandit problems [19, 25].

## 2.2 Thompson Sampling for Contextual Logistic Bandits

TS provides a flexible and computationally tractable framework for inference in contextual logistic bandits. TS for the contextual bandit is broadly defined in Bayesian terms, where a prior distribution $p(\boldsymbol{\theta})$ over the parameter $\boldsymbol{\theta}$ is updated iteratively using a set of historical observations $\mathcal{D}_{t-1} = \{(\mathbf{x}_{i,a_i}, a_i, r_i)|i = 1, \ldots, t-1\}$. The posterior distribution $p(\boldsymbol{\theta}|\mathcal{D}_{t-1})$ is calculated using Bayes' rule and is proportional to the distribution $\prod_{i=1}^{t-1} p(r_i|a_i, \mathbf{x}_{i,a_i}, \boldsymbol{\theta})p(\boldsymbol{\theta})$. A random sample $\boldsymbol{\theta}_t$ is drawn from this posterior, corresponding to a stochastic estimate of $\boldsymbol{\theta}^*$ after $t$ steps. The optimal arm is then the arm offering the highest reward with respect to the current estimate $\boldsymbol{\theta}_t$. In other words, the arm with the highest expected reward is chosen according to a probability $p(a_t = a|\boldsymbol{\theta}_t, \mathcal{D}_{t-1})$, which is expressed formally as

$$\int \mathbb{1}_{\mathcal{A}_t^{\max}(\boldsymbol{\theta}_t)} \left( E[r_t|a, \mathbf{x}_{t,a}, \boldsymbol{\theta}_t] \right) p(\boldsymbol{\theta}_t|\mathcal{D}_{t-1})d\boldsymbol{\theta}_t, \tag{2}$$

where $\mathcal{A}_t^{\max}(\boldsymbol{\theta}_t)$ is the set of arms with maximum rewards at step $t$ if the true parameter were $\boldsymbol{\theta}_t$.

After $t$ steps, the joint probability mass function over the rewards $r_1, r_2, \ldots, r_t$ observed upon taking actions $a_1, a_2, \ldots, a_t$ is $\prod_{i=1}^{t} p(r_i = 1|a_i, \mathbf{x}_{i,a}, \boldsymbol{\theta}_i)$ or

$$\prod_{i=1}^{t} \mu(\mathbf{x}_{i,a_i}^\top \boldsymbol{\theta}_i)^{r_i} [1 - \mu(\mathbf{x}_{i,a_i}^\top \boldsymbol{\theta}_i)]^{1-r_i}, \tag{3}$$

where $\boldsymbol{\theta}_1, \boldsymbol{\theta}_2, \ldots, \boldsymbol{\theta}_t$ are the estimates of $\boldsymbol{\theta}^*$ at each trial up to $t$.

In the case of logistic regression for binary rewards, the posterior derived from this joint probability is intractable. Laplace-TS addresses this issue by approximating the posterior with a multivariate Gaussian distribution with a diagonal covariance matrix following a Laplace approximation. The mean of this distribution is the *maximum a posteriori* estimate and the inverse variance of each feature is the curvature [15].

Laplace approximations are effective in finding smooth densities peaked around their posterior modes, and are thus applicable to the logistic posterior, which is strictly exp-concave [18]. This approach has

shown superior empirical performance versus UCB algorithms [9] and other TS-based approximation methods [30]. Laplace-TS is therefore an appropriate benchmark in the evaluation of contextual bandit algorithms using TS approaches.

## 3 Pólya-Gamma Augmentation for Logistic Contextual Bandits

The Laplace approximation leads to simple, iterative algorithms, which in the offline setting lead to accurate estimates across a potentially large number of sparse models [7]. In this section, we propose PG-TS, an alternative approach stemming from recent developments in augmentation for Bayesian inference in logit models [27, 31].

### 3.1 The Pólya-Gamma Augmentation Scheme

Consider a logit model with $t$ binary observations $r_i \sim Bin(1, \mu(\mathbf{x}_i^\top \boldsymbol{\theta}))$, parameter $\boldsymbol{\theta} \in \mathbb{R}^d$ and corresponding regressors $\mathbf{x}_i \in \mathbb{R}^d$, $i = 1, \ldots, t$. To estimate the posterior $p(\boldsymbol{\theta}|\mathcal{D}_t)$, classic MCMC methods use independent and identically distributed (i.i.d) samples. Such samples can be challenging to obtain, especially if the dimension $d$ is large [10]. To address this challenge, we reframe the discrete rewards as functions of latent variables with Pólya-Gamma (PG) distributions over a continuous space [27]. The PG latent variable construction relies on the theoretical properties of PG random variables to exploit the fact that the logistic likelihood is a mixture of Gaussians with PG mixing distributions [27, 12, 13].

**Definition 1** *Let $X$ be a real-valued random variable. $X$ follows a Pólya-Gamma distribution with parameters $b > 0$ and $c \in \mathbb{R}$, $X \sim PG(b, c)$ if the following holds:*

$$X = \frac{1}{2\pi^2} \sum_{k=1}^{\infty} \frac{G_k}{(k - 1/2)^2 + c^2/(4\pi^2)},$$

*where $G_k \sim Ga(b, 1)$ are independent gamma variables.*

The identity central to the PG augmentation scheme [27] is

$$\frac{(e^\psi)^a}{(1 + e^\psi)^b} = 2^{-b} e^{\kappa\psi} \int_0^\infty e^{-\omega\psi^2/2} p(\omega) d\omega, \tag{4}$$

where $\psi \in \mathbb{R}$, $a \in \mathbb{R}$, $b > 0$, $\kappa = a - b/2$ and $\omega \sim PG(b, 0)$. When $\psi = \mathbf{x}_t^\top \boldsymbol{\theta}$, the previous identity allows us to write the logistic likelihood contribution of step $t$ as

$$L_t(\boldsymbol{\theta}) = \frac{[\exp(\mathbf{x}_t^\top \boldsymbol{\theta})]^{r_t}}{1 + \exp(\mathbf{x}_t^\top \boldsymbol{\theta})} \propto \exp(\kappa_t \mathbf{x}_t^\top \boldsymbol{\theta}) \int_0^\infty \exp[-\omega_t (\mathbf{x}_t^\top \boldsymbol{\theta})^2 / 2] p(\omega_t; 1, 0) d\omega_t,$$

where $\kappa_t = r_t - 1/2$ and $p(\omega_t; 1, 0)$ is the density of a PG-distributed random variable with parameters $(1, 0)$. In turn, the conditional posterior of $\boldsymbol{\theta}$ given latent variables $\boldsymbol{\omega} = [\omega_1, \ldots, \omega_t]$ and past rewards $\mathbf{r} = [r_1, \ldots, r_t]$ is a conditional Gaussian:

$$p(\boldsymbol{\theta}|\boldsymbol{\omega}, \boldsymbol{r}) = p(\boldsymbol{\theta}) \prod_{i=1}^{t} L_i(\boldsymbol{\theta}|\omega_i) \propto p(\boldsymbol{\theta}) \prod_{i=1}^{t} \exp\{\frac{\omega_i}{2}(\mathbf{x}_i^\top \boldsymbol{\theta} - \kappa_i/\omega_i)^2\}.$$

With a multivariate Gaussian prior for $\boldsymbol{\theta} \sim MVN(\mathbf{b}, \mathbf{B})$, this identity leads to an efficient Gibbs sampler. The main parameters are drawn from a Gaussian distribution, which is parameterized with latent variables drawn from the PG distribution [27]. The two steps are:

$$(\omega_i|\boldsymbol{\theta}) \sim PG(1, \mathbf{x}_i^\top \boldsymbol{\theta}) \tag{5}$$
$$(\boldsymbol{\theta}|\mathbf{r}, \boldsymbol{\omega}) \sim \mathcal{N}(\mathbf{m}_\omega, \mathbf{V}_\omega), \tag{6}$$

with $\mathbf{V}_\omega = (\mathbf{X}^\top \boldsymbol{\Omega} \mathbf{X} + \mathbf{B}^{-1})^{-1}$, and $\mathbf{m}_\omega = \mathbf{V}_\omega (\mathbf{X}^\top \boldsymbol{\kappa} + \mathbf{B}^{-1} \mathbf{b})$ where $\boldsymbol{\kappa} = [\kappa_1, \ldots, \kappa_t]$.

Conveniently, efficient algorithms for sampling from the PG distribution exist [27]. Based on ideas from Devroye [12, 13], which avoid the need to truncate the infinite sum in Eq 4, the algorithm relies on an accept-reject strategy for which the proposal distribution only requires exponential, uniform,

and Gaussian random variables. With an acceptance probability uniformly lower bounded by 0.9992 (at most 9 rejected draws out of every $10,000$ proposed), the resulting algorithm is more efficient than all previously proposed augmentation schemes in terms of both effective sample size and effective sampling rate [27]. Furthermore, the PG sampling procedure leads to a uniformly ergodic mixture transition distribution of the iterative estimates $\{\boldsymbol{\theta}_i\}_{i=0}^{\infty}$ [10]. This result guarantees the existence of central limit theorems regarding sample averages involving $\{\boldsymbol{\theta}_i\}_{i=0}^{\infty}$ and allows for consistent estimators of the asymptotic variance. The advantage of PG augmentation has been proven in multiple Gibbs sampling and variational inference approaches, including binomial models [27], multinomial models [24], and negative binomial regression models with logit link functions [38, 31]. In the next section, we leverage its strengths to perform online, fully Bayesian inference for logistic contextual bandits with state-of-the-art performance.

## 3.2 PG-TS Algorithm Definition

Our algorithm, PG-TS, uses the PG augmentation scheme to represent the binomial distributions of the sequential rewards in terms of latent variables with Gaussian distributions to perform tractable Bayesian logistic regression in a Thompson sampling setting.

We consider a multivariate Gaussian distribution over parameter $\boldsymbol{\theta} \sim MVN(\mathbf{b}, \mathbf{B})$ with prior mean $\mathbf{b}$ and covariance $\mathbf{B}$. For simplicity, let $\mathbf{X}_t$ be the $d \times t$ design matrix $[\mathbf{x}_1, \ldots, \mathbf{x}_t]$ that includes the context of all arms chosen up to round $t$. $\boldsymbol{\Omega}_t$ is the diagonal matrix of the PG auxiliary variables $[\omega_1, \ldots, \omega_t]$ and let $\boldsymbol{\kappa}_t = [r_1 - \frac{1}{2}, \ldots, r_t - \frac{1}{2}]$. Further, let $\boldsymbol{r}_t = [r_1, \ldots, r_t]$ be the history of rewards.

The PG-TS algorithm uses a Gibbs sampler based on the PG augmentation scheme to approximate the logistic likelihood corresponding to observations up to the current step. At each step, sampling from the posterior is exact. The ergodicity of the sampler guarantees that, as the number of trials increases, the algorithm is able to consistently estimate both the mean and the variance of parameter $\boldsymbol{\theta}$ [36].

We sample from the PG distribution [24, 27] including $M = 100$ burn-in steps. This number is empirically tuned, as evaluating how close a sampled $\boldsymbol{\theta}_t$ is to the true GLM estimator $\boldsymbol{\theta}_t^{GLM}$ as a function of the burn-in step $M$ is a challenging problem. Thus, frequentist-derived TS algorithms and regret bounds cannot be derived for the PG distributions, unlike other formulations of this problem [2]. In our empirical studies, we find PG-TS with $M = 100$ to be sufficient for reliable mixing, as measured by the competitive regret achieved. When $M = 1$, the resulting algorithm, PG-TS-stream, is reminiscent of a streaming Gibbs inference scheme. In practice, this leads to a faster algorithm. As shown in the Results, PG-TS-stream shows competitive performance in terms of cumulative rewards in both simulated and real-world data scenarios.

---

**Algorithm 1** PG-TS

**Input:** $\mathbf{b}, \mathbf{B}, M, \mathcal{D} = \emptyset, \boldsymbol{\theta}_0 \sim MVN(\mathbf{b}, \mathbf{B})$
**for** $t = 1, 2, \ldots$ **do**
    Receive contexts $\mathbf{x}_{t,a} \in \mathbb{R}^d$
    $\boldsymbol{\theta}_t^{(0)} \leftarrow \boldsymbol{\theta}_{t-1}$
    **for** $m = 1$ **to** $M$ **do**
        **for** $i = 1$ **to** $t - 1$ **do**
            $\omega_i | \boldsymbol{\theta}_t^{(m-1)} \sim PG(1, \mathbf{x}_{i,a_i}^{\top} \boldsymbol{\theta}_t^{(m-1)})$
        $\boldsymbol{\Omega}_{t-1} = diag(\omega_1, \omega_2, \ldots \omega_{t-1})$
        $\boldsymbol{\kappa}_{t-1} = \left[r_1 - \frac{1}{2}, \ldots, r_{t-1} - \frac{1}{2}\right]^{\top}$
        $\mathbf{V}_\omega \leftarrow (\mathbf{X}_{t-1}^{\top} \boldsymbol{\Omega}_{t-1} \mathbf{X}_{t-1} + \mathbf{B}^{-1})^{-1}$
        $\mathbf{m}_\omega \leftarrow \mathbf{V}_\omega (\mathbf{X}^{\top} \boldsymbol{\kappa}_{t-1} + \mathbf{B}^{-1}\mathbf{b})$
        $\boldsymbol{\theta}_t^{(m)} | \boldsymbol{r}_{t-1}, \boldsymbol{\omega} \sim MVN(\mathbf{m}_\omega, \mathbf{V}_\omega)$
    $\boldsymbol{\theta}_t \leftarrow \boldsymbol{\theta}_t^{(M)}$
    Select arm $a_t \leftarrow argmax_a \mu(\mathbf{x}_{t,a}^{\top} \boldsymbol{\theta}_t)$
    Observe reward $r_t \in \{0, 1\}$
    $\mathcal{D} = \mathcal{D} \cup \{\mathbf{x}_{t,a_t}, a_t, r_t\}$

---

# 4 Results of PG-TS for contextual bandit applications

We evaluate and compare our PG-TS method with Laplace-TS. Laplace-TS has been shown to outperform its UCB competitors in all settings considered here [9].

We evaluate our algorithm in three scenarios: simulated data sets with parameters sampled from Gaussian and mixed Gaussian distributions, a toy data set based on the Forest Cover Type data set from the UCI repository [15], and an offline evaluation method for bandit algorithms that relies on real-world log data [23].

## 4.1 Generating Simulated Data

**Gaussian simulation.** We generated a simulated data set with 100 arms and 10 features per context across $1,000$ trials. We generated contexts $\mathbf{x}_{t,a} \in \mathbb{R}^{10}$ from multivariate Gaussian distributions $\mathbf{x}_{t,a} \sim MVN(-\mathbf{3}, \mathbf{I}_{10})$ for all arms $a$. The true parameters were simulated from a multivariate Gaussian with mean 0 and identity covariance matrix, $\boldsymbol{\theta}^* \sim MVN(\mathbf{0}, \mathbf{I}_{10})$. The resulting reward associated with the optimal arm was 0.994 and the mean reward was 0.195. We set the hyperparameters $\mathbf{b} = \mathbf{0}$, and $\mathbf{B} = \mathbf{I}_{10}$. We averaged the experiments over 100 runs.

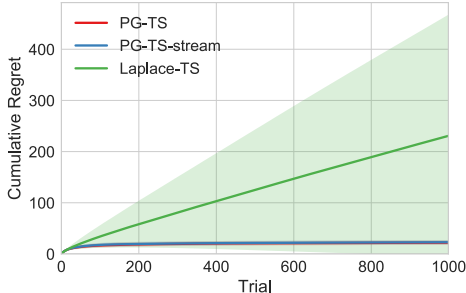

Figure 1: Comparison of the average cumulative regret of the PG-TS, PG-TS-stream, and Laplace-TS algorithms on the simulated data set with Gaussian $\boldsymbol{\theta}^*$ over 100 runs with $1,000$ trials (standard deviation shown as shaded region). Both PG-TS and PG-TS-stream outperform Laplace-TS in consistently achieving lower cumulative regret.

We first considered the effect of the burn-in parameter $M$ on the resulting average cumulative regret (Eq. 1; Fig. S1 Supplementary Material). As expected, larger $M$ led to lower regret, as the Markov chain had more time to mix. We note that $M > 100$ burn-in iterations was not noticeably better than $M = 100$, while the computational time grew. Interestingly, the average cumulative regret of PG-TS-stream with $M = 1$ was comparable to that of PG-TS. This suggests that, after a number of steps greater than the number of iterations necessary for mixing, the sampler in PG-TS-stream has had sufficient time to mix.

In this simulation, both PG-TS strategies outperformed their Laplace counterpart, which failed to converge on average (Fig. 1). This behavior is expected: due to its simple Gaussian approximation, Laplace-TS does not always converge to the global optimum of the logistic likelihood in non-asymptotic settings.

Furthermore, the PG-TS algorithms outperform Laplace-TS in terms of balancing exploration and exploitation: Laplace-TS gets stuck on sub-optimal arm choices, while PG-TS continues to explore relative to the estimated variance of the posterior distribution of $\boldsymbol{\theta}$ to find the optimal arm (Fig. 2).

**Mixture of Gaussians: Prior misspecification.** Laplace approximations are sensitive to multimodality. We therefore explored a prior misspecification scenario, where true parameter $\boldsymbol{\theta}^*$ is sampled from a four-component Gaussian mixture model, as opposed to the Gaussian distribution assumed by both algorithms. As before, we simulated a data set with 100 arms, each with 10 features, and marginally independent contexts $\mathbf{x}_{t,a} \sim MVN(\mathbf{0}, \mathbf{I}_{10})$, across $5,000$ trials.

The true parameters were generated from a mixed model with variances $\sigma^2_{j=1:4} \sim$ Inverse-Gamma$(3, 1)$, means $\mu_{j=1:4} \sim N(-3, \sigma^2_j)$, and mixture weights $\phi \sim$ Dirichlet$(1, 3, 5, 7)$ such that $\theta^*(i) \sim \sum_{j=1}^4 \phi_j N(\mu_j, \sigma^2_j)$, with $\boldsymbol{\theta}^* = [\theta^*_1, \theta^*_2, \ldots, \theta^*_{10}]$. The reward associated with the optimal arm was 0.999 and the mean reward was 0.306. We found that the misspecified model does not prevent the PG-TS algorithms from consistently finding the correct arm, while Laplace-TS exhibits poor average behavior (Fig. S3 Supplementary Materials).

## 4.2 PG-TS applied to Forest Cover Type Data

We further compared these methods using the Forest Cover Type data from the UCI Machine Learning repository [8]. These data contain $581,021$ labeled observations from regions of a forest area. The labels indicate the dominant species of trees (cover type) in each region. Following the preprocessing pipeline proposed by [15], we centered and standarized the 10 non-categorical variables and added a constant covariate; we then partitioned the $581,012$ samples into $k = 32$ clusters using unsupervised mini-batch $k$-means clustering. We took the cluster centroids to be the contexts corresponding to each of our arms. To fit the logistic reward model, rewards were binarized for each data point by associating the first class "Spruce/Fir" to a reward of 1, and to a reward of 0 otherwise. We then set the reward for each arm to be the average reward of the data points in the corresponding cluster; these ranged from 0.020 to 0.579. The task then becomes the problem of finding the cluster with the highest

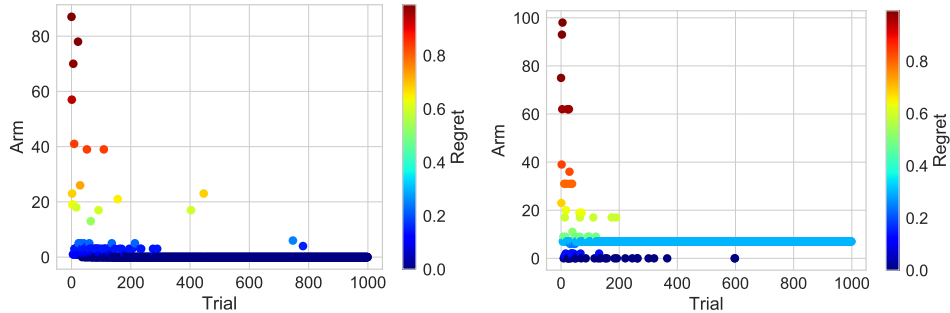

Figure 2: Comparison of arm choices for the PG-TS (Left) and Laplace-TS algorithms (Right) on simulated data with Gaussian $\boldsymbol{\theta}^*$ across $1,000$ trials. Arms were sorted by expected reward in decreasing order, with arm 0 giving the highest reward, and arm 99 the lowest. The selected arms are colored by the distance of their expected reward from the optimal reward (regret). Laplace-TS gets stuck on a sub-optimal arm, while PG-TS explores successfully and settles on the optimal one.

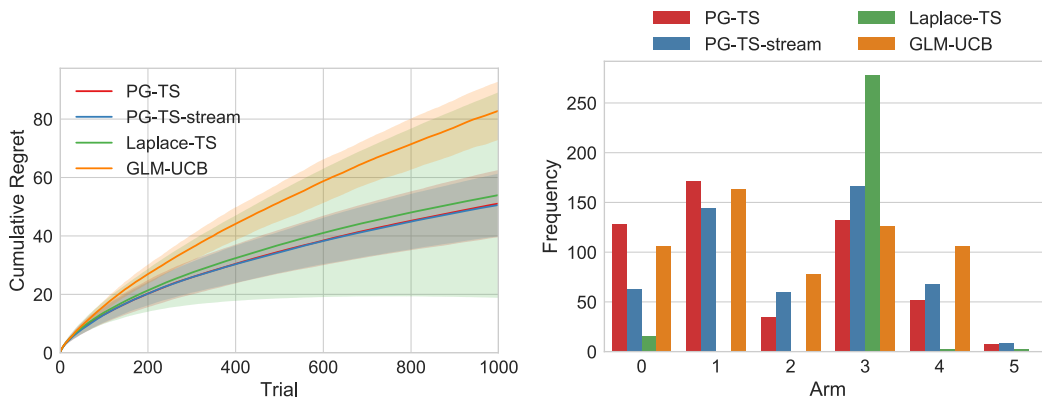

Figure 3: Left: comparison of the average cumulative regret of the PG-TS, PG-TS-stream, Laplace-TS, and GLM-UCB algorithms on the Forest Cover Type data over $100$ runs with $1,000$ trials (one standard deviation shaded). PG-TS significantly outperforms Laplace-TS and GLM-UCB, with slight improvement over PG-TS-stream. Right: median frequencies of the six best arms' draws. The arms were sorted by expected reward in decreasing order, with arm 0 giving the highest reward, and arm 5 the lowest. PG-TS explores better than Laplace-TS, which gets stuck in a sub-optimal arm.

proportion of Spruce/Fir forest cover in a setting with 32 arms and 11 context features. As a baseline, we implemented the generalized linear model upper confidence bound algorithm (GLM-UCB) [15].

On this forest cover task, the PG-TS algorithms show improved cumulative regret with respect to both the Laplace-TS and the GLM-UCB procedures, with PG-TS performing slightly better of the two (Fig. 3). Both PG-TS and PG-TS-stream explored the arm space more successfully, and exploited high-reward arms with a higher frequency than their competitors (Fig. 3).

### 4.3 PG-TS Applied to News Article Recommendation

We evaluated the performance of PG-TS in the context of news article recommendations on the public benchmark Yahoo! Today Module data through an unbiased offline evaluation protocol [22]. As before, users are assumed to click on articles in an i.i.d manner. Available articles represent the pool of arms, the binary payoff is whether a user clicks on a recommended article, and the expected payoff of an article is the *click-through rate* (CTR). Our goal is to choose the article with the maximum expected CTR at each visit, which is equivalent to maximizing the total expected reward. The full data set contains $45,811,883$ user visits from the first 10 days of May 2009; for each user visit, the module features one article from a changing pool of $K \approx 20$ articles, which the user either clicks ($r = 1$) or does not click ($r = 0$). We use $200,000$ of these events in our evaluation for efficiency; $\leq 24,000$

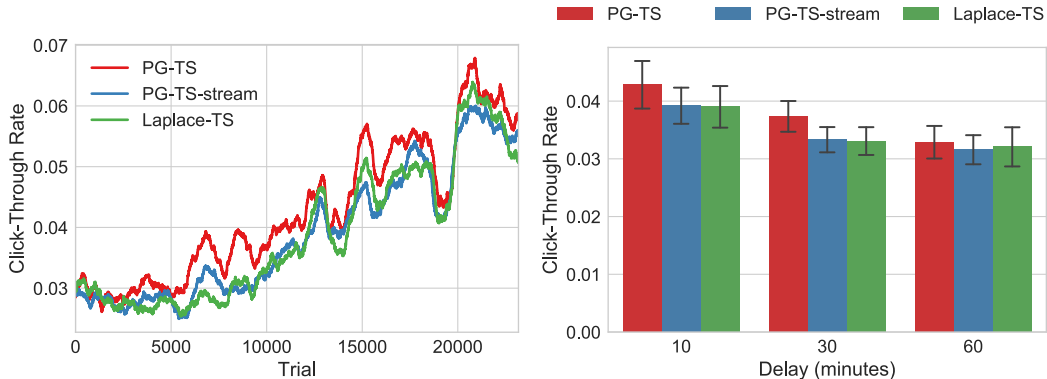

Figure 4: Comparison of the average click-through rate (CTR) achieved by the PG-TS, PG-TS-stream, and Laplace-TS algorithms with 10-minute delay (Left) and with varying delay (Right) on $24,000$ events in the Yahoo! Today Module data set over 20 runs. Left: the moving average CTR is observed every $1,000$ observations. Right: the standard deviation of the average CTR is shown. PG-TS achieves higher CTR across all delays, especially for short delays.

of these are valid events for each of our evaluated algorithms. Each article is associated with a feature vector (context) $\mathbf{x} \in \mathbb{R}^6$ including a constant feature capturing an intercept, preprocessed using a conjoint analysis with a bilinear model [11]; note that we do not use user features as context. In this evaluation, we maintain separate estimates $\boldsymbol{\theta}_a$ for each arm. We also update the model in batches (groups of observations across time *delays*) to better match the real-world scenario where computation is expensive and delay is necessary. In all settings, PG-TS consistently and significantly out-performs the Laplace-TS approach (Fig. 4). In particular, PG-TS shows a significant improvement in CTR across all delays. Note that PG-TS benefits in performance in particular with short delays. Despite showing only marginal improvement when compared to Laplace-TS, PG-TS-stream offers the advantage of a flexible, fast data streaming approach without compromising performance on this task.

## 5 Discussion

We introduced PG-TS, a fully Bayesian algorithm based on the Pólya-Gamma augmentation scheme for contextual bandits with logistic rewards. This is the first method where Pólya-Gamma augmentation is leveraged to improve bandit performance. Our approach addresses two deficiencies in current methods. First, PG-TS provides an efficient online approximation scheme for the analytically intractable logistic posterior. Second, because PG-TS explicitly estimates context feature covariances, it is more effective in balancing exploration and exploitation relative to Laplace-TS, which assumes independence of each context feature. We showed through extensive evaluation in both simulated and real-world data that our approach offers improved empirical performance while maintaining comparable computational costs by leveraging the simplicity of the Thompson sampling framework. We plan to extend our framework to address computational challenges in high-dimensional data via hash-amenable extensions [20].

Motivated by our results and by recent work on PG inference in dependent multinomial models [24], we aim to extend our work to the problem of multi-armed bandits with categorical rewards. We further envision a generalization of this approach to sampling in bandit problems where additional structure is imposed on the contexts; for example, settings where arm contexts are sampled from dynamic linear topic models [17], or settings in which social network information is available for users and contexts [16].

Future work will address the more general reinforcement learning setting of Bayes-Adaptive MDP with discrete state and action sets [14]. In this case, the state transition probabilities are multinomial distributions; therefore, our online Pólya-Gamma Gibbs sampling procedure can be extended to approximate the emerging intractable posteriors.

**Acknowledgments**

We would like to thank Scott Linderman, Diana Cai, and Jean Feng for insightful discussions and their helpful feedback. Lastly, we thank all the anonymous reviewers for their valuable comments.

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
