[Supplementary Material · SupplementaryMaterials.pdf]

# Supplementary Figures for PG-TS: Improved Thompson Sampling for Logistic Contextual Bandits

## The effect of the burn-in step $M$ in Gaussian Simulations

PG-TS relies on approximating an integral using a double sampling of an appropriate Markov Chain. Hence, the burn-in Gibbs step $M$ affects the convergence behavior of PG-TS with respect to the cumulative regret (Fig. S1). Due to diminishing returns illustrated in our empirical studies, we set $M$ to 100 throughout the paper. In practice, setting $M$ to a large value allows for appropriate mixing and has substantial impact on performance, as seen in the news recommendation application.

Figure S1: Comparison of the average cumulative regret of the PG-TS algorithm with varying number of burn-in iterations on the simulated data set with Gaussian $\boldsymbol{\theta}^*$ over 100 runs with $1,000$ trials. The lower the regret, the better the performance.

## Variance of the Cumulative Regret Performance

The methods considered show very diverse behavior across experiments even in the simple Gaussian simulation case. In particular, while both PG-TS and PG-TS-stream converge across experiments, Laplace-TS shows high variability and significantly higher cumulative regret across the same trials.

Figure S2: Trace plots of cumulative regret for PG-TS and PG-TS-stream (Top), and Laplace-TS (Bottom) on the simulated data set with Gaussian $\boldsymbol{\theta}^*$ over 100 runs with $1,000$ trials.

Furthermore, Laplace-TS is sensitive to multimodality. We found that the misspecified model does not prevent the PG-TS algorithms from consistently finding the correct arm, while Laplace-TS exhibits poor average behavior (Fig. S3). For our similations, we do not show comparison to GLM-UCB as previous studies address the superiority of Laplace-TS [Chapelle and Li, 2011, Russo and Van Roy, 2014].

Figure S3: Comparison of the average cumulative regret of the PG-TS, PG-TS-stream, and the Laplace-TS algorithms on simulated data with mixed Gaussian $\theta^*$ over 100 runs with 5,000 trials (standard deviation shaded). Laplace-TS performs better during earlier trials, yet struggles to settle on an optimal arm.

## Langevin Alternatives

We compared our method to Langevin-TS [Russo et al., 2017], and we found that PG-TS significantly outperforms Langevin-TS in our simulations (Fig. S5). We note that the Langevin implementation is very sensitive to learning rate, step size and numerous other initialization parameters, unlike PG-TS whose performance is consistent across our simulations.

Figure S4: Comparison of the average cumulative regret of the PG-TS-iter, PG-TS-stream, and Laplace-TS algorithms and Langevin on the simulated data set with Gaussian $\theta^*$ over 100 runs with 1,000 trials (standard deviation shown as shaded region)

We further note that PG-TS outperforms a variance reduced stochastic gradient Monte Carlo approach extension to Langevin-TS [Chatterji et al., 2018].

Figure S5: Comparison of the average cumulative regret of the PG-TS-iter, PG-TS-stream, and Laplace-TS algorithms and Langevin on the simulated data set with Gaussian $\theta^*$ over 100 runs with $1,000$ trials (standard deviation shown as shaded region)

## Exploration and Exploitation comparison across Gaussian simulations

Figure S6: Comparison of the arm choices for the GLM-UCB (Left) and PG-TS-stream (Right) algorithms on the simulated data set with Gaussian $\theta^*$ across $1,000$ trials. The arms were sorted by expected reward in decreasing order, with arm 0 giving the highest reward, and arm 99 the lowest. The selected arms are colored according to the distance of their expected reward from the optimal reward (regret). GLM-UCB takes many trials to settle on the optimal arm, while both PG-TS algorithms explore successfully and settle on the optimal one. Recall that Laplace-TS gets stuck on a sub-optimal arm.

## Pseudocode for the algorithms mentioned

---

**Algorithm 1** Generic Contextual Bandit Algorithm

---

**Initialize** $\mathcal{D}_0 = \emptyset$
**for** $t = 1, 2, ...$ **do**
    Observe $K_t$ arms $\mathcal{A}_t$
    Receive context $\mathbf{x}_{t,a} \in \mathbb{R}^d$
    Select $a_t$ given $\mathbf{x}_{t,a}, \mathcal{D}_{t-1}$
    Observe reward $r_{t,a_t}$
    Update $\mathcal{D}_t = \mathcal{D}_{t-1} \cup \{\mathbf{x}_{t,a_t}, a_t, r_t\}$
**end for**

---

---

**Algorithm 2** Laplace-TS [Chapelle and Li, 2011]

---

**Input:** Regularization parameter $\lambda = 1$
$m_i = 0, q_i = \lambda$, for $i = 1, 2, \ldots d$
**for** $t = 1$ **to** $T$ **do**
    Receive context $\mathbf{x}_{t,a}$
    $\mathbf{Q} = diag\left(q_1^{-1}, q_2^{-1}, \ldots q_d^{-1}\right)$
    Draw $\boldsymbol{\theta}_t \sim MVN\left(\mathbf{m}, \mathbf{Q}\right)$
    Select $a_t = \arg\max_a \mu\left(\mathbf{x}_{t,a}^\top \boldsymbol{\theta}_t\right)$
    Receive reward $r_t$
    $y_t = 2r_t - 1$
    $\mathbf{w} = \arg\min_{\mathbf{w}} \frac{1}{2} \sum_{i=1}^d q_i \left(w_i - m_i\right)^2 - \log\left(\mu\left(y_t \mathbf{x}_{t,a_t}^\top \mathbf{w}\right)\right)$
    $\mathbf{m} = \mathbf{w}$
    $p = \mu\left(\mathbf{x}_{t,a_t}^\top \mathbf{w}\right)$
    $\mathbf{q} = \mathbf{q} + p\left(1 - p\right)\mathbf{x}_{t,a_t}^2$
**end for**

---

---

**Algorithm 3** GLM-UCB [Filippi et al., 2010]

---

**Input:** Admissible parameter set $\boldsymbol{\Theta}$, slowly increasing function $\rho(t)$
**for** $t = 1, 2, \ldots$ **do**
    Receive context $\mathbf{x}_{t,a}$
    $\boldsymbol{\theta}_t = \arg\min_{\theta \in \boldsymbol{\Theta}} \left\|\sum_{i=1}^{t-1}(r_i - \mu(\mathbf{x}_{i,a_i}^\top \boldsymbol{\theta}))\mathbf{x}_{i,a_i}\right\|_{\mathbf{V}_t^{-1}}^2$
    Select $a_t = \arg\max_a \{x_{t,a}^\top \boldsymbol{\theta}_t + \rho(t) \|x_{t,a}\|_{\mathbf{V}_t^{-1}}^2\}$
    Receive reward $r_t \in \{0, 1\}$
    $\mathbf{V}_{t+1} = \sum_{i=1}^t \mathbf{x}_{i,a_i} \mathbf{x}_{i,a_i}^\top$
**end for**

---