[Reviews · NeurIPS 2018]

Reviewer 1



The paper presents Polya Gamma augmentation of Thompson Sampling for contextual bandits providing an alternative to Laplace approximation, an as existing Bayesian solution. The paper presents extensive experimental results considering both synthetic and real data to demonstrate PG-TS's superiority over Laplace-TS. Quality: The paper is of high quality. Clarity: The paper is well-written with a good coverage of relevant literature, motivation of the problem, clear algorithm description, reproducible experimental setup. Originality: The paper addresses an important problem and provides a novel solution. Significance: The paper addresses a relevant problem and presents a novel approach that works well in practice. Question: Under what condition (if any) Laplace-TS can possibly outperform PG-TS?

Reviewer 2



Summary: The authors address the important problem of efficient Thompson sampling (1) for regret minimization in logistic contextual bandits. While UCB and its various extensions have been analyzed in the literature, Thompson sampling has been shown to give good empirical performance in real-world settings. The typical Laplace approximation that is employed for sampling from the posterior distribution can lead to suboptimal arms is shown via simulations and real-world experiments. They ameliorate the situation by proposing to use Polya-Gamma augmented sampling which was recently introduced for Bayesian inference in logistic models (Polson et al 2013). Comments: The problem is well-defined and the presentation is very clear with links to the relevant literature. The main contribution is the application of Polya-Gamma distributed variables for the latent variables of the Gaussian distributed parameter vector Theta to the problem of logistic contextual bandits. The experiments are convincing by using both the synthetic examples and real-world experiments. Overall a good application paper and could have been strengthened if any of the next steps highlighted in the discussion section such as multinomial models were also tackled. My concern is the heavy reliance on previous work for improving the posterior sampling step and if it is offset by the improved results shown when applied to bandits. (1) Daniel J. Russo, Benjamin Van Roy, Abbas Kazerouni, Ian Osband and Zheng Wen (2018), “A Tutorial on Thompson Sampling”, Foundations and TrendsR in Machine Learning: Vol. 11, No. 1, pp 1–96. DOI: 10.1561/2200000070

Reviewer 3



This submission tries to propose Polya-gamma augmented Thompson sampling from a frequentist-derived algorithm whose likelihood is a mixture of Gaussian and PG mixing distribution when the reward is the logistic model with binary observation in contextual bandits to improve the Laplace approximation based Bayesian sampling. Experiments are demonstrated on the real world data including the public benchmark Yahoo data. Detailed comments: Like you admitted the logistic reward is pragmatic, that's why so far very few have been researched on this topic, which makes it difficult to get adopted by the community. As you stated the example in the news recommendation setting case in the click or not over the item problem, which is the same or similar to recent advances that took over the contextual bandit, see below. You adopted the traditional Gibbs sampler, where you sample from the PG distribution with M = 100 burn-in steps, for the experiment is fine, but lack of regret bounds to support, that you may want to study more, e.g., see [1]'s section or appendix about TS. Laplace-TS's batch update is a good pro but did not fully explored at the moment. Gibbs step M is too tiny and does not well theoretically motivated. Laplace approximations are sensitive to multimodality is a con. The sample complexity or size etc would be clearly stated and analyzed. You played with a toy dataset on Forest Cover Type from UCI where you only reported the result over 100 runs with 1000 trials, I suggest adjusting the performance towards at least, say, 50k or 100k etc. There is no way to adopt some wrong results as illustrated in Figure 4, it may take a while for you to conquer though. Your major issue is lack of theoretical guarantee or regret (either upper or lower) bound for the proposed algorithm, as it's standard in this field. The regret gain is not significant from Figure 5 through the experimental report presented, and the benefit of arm's frequency is not conniving as far as I can tell. You partitioned the UCI dataset to k = 32 clusters by the traditional k-means clustering where you set the reward of each arm to be the average reward in the corresponding cluster, where you might want to change k's values to see the difference. Missing references: [1] ICML 2017 On Context-Dependent Clustering of Bandits [2] SIGIR 2016 Collaborative Filtering Bandits [3] ICML 2016 Distributed Clustering of Linear Bandits in Peer to Peer Networks Where the exactly same Yahoo today module benchmark data is employed, in this sense that your proposed solutions should be able to compare with the state-of-the-art overtook traditional contextual bandit[1-3], which I am curious to see the empirical comparison at least with either one of [1-3]. Since there is no regret bound provided in this manuscript, the empirical result as described and in Figure 4 which showed that your result is wrong or inconsistent on your implementation for conducting the experiment on Yahoo, see more details on how to do legitimate experiments therein in [2,3] to observe the correct performance behavior you should achieve. The discussion is not really a discussion part, and the real conclusion is missing. Please differentiate these two sections and wrote them separately. Last but not least, this manuscript should be written in a more professional English writing including to fix typos and grammar errors etc, e.g., "Motivated by our results" and so forth. You should carefully address these concerns to increase the quality of this submission. %Update after rebuttal% I read their rebuttal, and they did a good job on the clarifications. I have increased my score, and I have no problem to accept this submission.